# Semi-supervised semantic segmentation needs strong, high-dimensional perturbations

## Abstract

Consistency regularization describes a class of approaches that have yielded ground breaking results in semi-supervised classification problems. Prior work has established the cluster assumption — under which the data distribution consists of uniform class clusters of samples separated by low density regions — as key to its success. We analyze the problem of semantic segmentation and find that the data distribution does not exhibit low density regions separating classes and offer this as an explanation for why semi-supervised segmentation is a challenging problem. We then identify the conditions that allow consistency regularization to work even without such low-density regions. This allows us to generalize the recently proposed CutMix augmentation technique to a powerful masked variant, CowMix, leading to a successful application of consistency regularization in the semi-supervised semantic segmentation setting and reaching state-of-the-art results in several standard datasets.

## 1 Introduction

Semi-supervised learning offers the tantalizing promise of training a machine learning model with limited amounts of labelled training data and large quantities of unlabelled data. These situations often arise in practical computer vision problems where large quantities of images are readily available and generating ground truth labels acts as a bottleneck due to the cost and labour required.

Consistency regularization (Sajjadi et al., 2016b; Laine & Aila, 2017; Miyato et al., 2017; Oliver et al., 2018) describes a class of semi-supervised learning algorithms that have yielded state-of-the-art results in semi-supervised classification, while being conceptually simple and often easy to implement. The key idea is to encourage the network to give consistent predictions for unlabeled inputs that are perturbed in various ways.

The effectiveness of consistency regularization is often attributed to the *smoothness assumption* (Miyato et al., 2017; Luo et al., 2018) or *cluster assumption* (Chapelle & Zien, 2005; Sajjadi et al., 2016a; Shu et al., 2018; Verma et al., 2019). The smoothness assumption states that samples close to each other are likely to have the same label. The cluster assumption — a special case of the smoothness assumption — states that decision surfaces should lie in low density regions, not crossing high density regions. This typically holds in classification tasks, where most successes of consistency regularization have been reported so far.

On a high level, semantic segmentation is classification, where each pixel is classified based on its neighborhood. It is therefore intriguing that consistency regularization has not been clearly beneficial in this context. We make the observation that the distance between patches centered on neighboring pixels varies smoothly even when the class of the center pixel changes, and thus there are no low-density regions on class boundaries. This alarming observation leads us to investigate the conditions that can allow consistency regularization to operate even in these conditions. Our key insight is that this is indeed possible, and that previous attempts have yielded little success primarily because the perturbations they used were not strong enough, and especially not high-dimensional enough.

We show that a flexibly masked variant of CutMix (Yun et al., 2019), which we call CowMix based on the mask appearance, does realize significant gains in semi-supervised semantic segmentation. This result clearly signposts a direction where further improvements are likely to be available.

## 2 BACKGROUND

Our work relates to prior art in three areas: recent regularization techniques for classification, semi-supervised classification with a focus on consistency regularization, and semantic segmentation.

### 2.1 MIXUP, CUTOUT, AND CUTMIX

The MixUp regularizer of Zhang et al. (2018) improves the performance of supervised image, speech and tabular data classifiers by using interpolated samples during training. The inputs and target labels of two randomly chosen examples are blended using the same randomly chosen factor.

The Cutout regularizer of DeVries & Taylor (2017) augments an image by masking a rectangular region to zero. The recently proposed CutMix regularizer of Yun et al. (2019) combines aspects of MixUp and Cutout, cutting a rectangular region from $B$ and pasting it over $A$. MixUp, Cutout, and CutMix improve supervised classification performance, with CutMix outperforming the other two.

### 2.2 SEMI-SUPERVISED CLASSIFICATION

The $\Pi$-model of Laine & Aila (2017) passes each unlabeled sample through a classifier twice, applying two realizations of a stochastic perturbation process, and minimizes the difference between the resulting class probability predictions. Their temporal model maintains a per-sample moving average of historical predictions and encourages subsequent predictions to be consistent with the average. Sajjadi et al. (2016b) similarly encourage consistency between the current and historical predictions. Miyato et al. (2017) improve the results by replacing the stochastic perturbations with adversarial directions, thus focusing on perturbations that are closer to the decision boundaries.

The mean teacher model of Tarvainen & Valpola (2017) encourages consistency between predictions of a student network and a teacher network. The teacher's weights are an exponential moving average (Polyak & Juditsky, 1992) of those of the student, leading to a faster convergence and improved results. French et al. (2018) adapt the mean teacher approach for domain adaptation.

Interpolation consistency training (ICT) (Verma et al., 2019) and MixMatch (Berthelot et al., 2019) both combine MixUp (Zhang et al., 2018) with consistency regularization. ICT uses the mean teacher model and applies MixUp to unsupervised samples, blending input images along with teacher class predictions to produce a blended input and target to train the student. MixMatch stochastically perturbs each sample multiple times and averages the predictions to produce unsupervised targets; MixUp is applied to labeled as well as unlabeled samples.

### 2.3 SEMANTIC SEGMENTATION

Long et al. (2015) finetune a pre-trained VGG-16 (Simonyan & Zisserman, 2014) image classifier to produce a dense set of predictions for overlapping input windows. They effectively transform the image classifier into a fully convolutional network that can be used to segment input images of arbitrary size. Various methods were proposed for increasing the localization accuracy of the results (Long et al., 2015; Chen et al., 2014; Mostajabi et al., 2014) until the introduction of encoder-decoder networks (Badrinarayanan et al., 2015; Ronneberger et al., 2015) led to a solution where the output resolution natively matches the input. In these recent methods the encoder downsamples the input progressively, similarly to image classifiers, the decoder performs progressive upsampling, and skip connections route data between the matching resolutions of the two networks, improving the ability to accurately segment fine details.

A number of approaches for semi-supervised semantic segmentation use additional data. Kalluri et al. (2018) use data from two datasets from different domains, maximizing the similarity between per-class embeddings from each dataset. Stekovic et al. (2018) use depth images and enforced geometric constraints between multiple views of a 3D scene. Relatively few approaches operate in a strictly semi-supervised setting. Hung et al. (2018) employ adversarial learning, using a discriminator network that distinguishes real from predicted segmentation maps to guide learning. Perone & Cohen-Adad (2018) apply consistency regularization to a MRI volume dataset and their method is the only successful application of consistency regularization to segmentation that we are aware of.

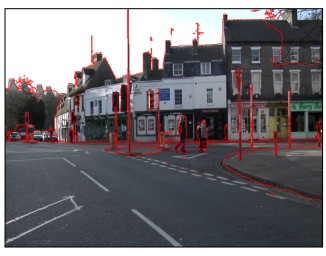
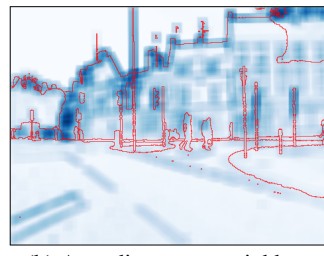
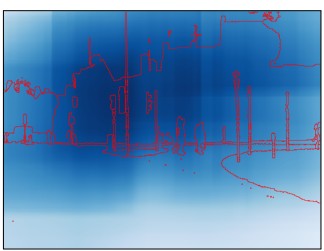

| (a) Example image | (b) Avg. distance to neighbor, patch size 15×15 | (c) Avg. distance to neighbor, patch size 225×225 |

Figure 1: In a segmentation task, low-density regions rarely correspond to class boundaries. (a) An image from Camvid dataset. (b) Average $L^2$ distance between raw pixel contents of a patch centered at pixel $p$ and four overlapping patches centred on the immediate neighbors of $p$, using 15×15 pixel patches. (c) Same for a more realistic receptive field size of 225×225 pixels. Dark blue indicates large inter-patch distance and therefore a low density region, white indicates a distance of 0. The red lines indicate segmentation ground truth boundaries.

## 3 CONSISTENCY REGULARIZATION FOR SEMANTIC SEGMENTATION

Consistency regularization adds a consistency loss term $L_{cons}$ to the loss that is minimized during training (Oliver et al., 2018). In a classification task, $L_{cons}$ measures a distance $d(\cdot, \cdot)$ between the predictions resulting from applying a neural network $f_\theta$ to an unsupervised sample $x$ and a perturbed version $\hat{x}$ of the same sample, i.e., $L_{cons} = d(f_\theta(x), f_\theta(\hat{x}))$. The perturbation used to generate $\hat{x}$ depends on the variant of consistency regularization used. A variety of distance measures $d(\cdot, \cdot)$ have been used, e.g., squared distance (Laine & Aila, 2017) or cross-entropy (Miyato et al., 2017).

Athiwaratkun et al. (2019) analyze a simplified version of the Π-model (Laine & Aila, 2017) in which perturbation consists of additive Gaussian noise so that $\hat{x} = x + \epsilon h$, where $h \sim \mathcal{N}(0, I)$ and $d(\cdot, \cdot)$ is a squared Euclidean distance. For small constant $\epsilon$, the expected value of the consistency loss term $L_{cons}$ is approximately proportional to the square of the Frobenius norm of the Jacobian $J_{f_\theta}(x)$ of the networks outputs with respect to its inputs:

$$\mathbb{E}[L_{cons}] = \mathbb{E}\big[\big\|f_\theta(x + \epsilon h) - f_\theta(x)\big\|^2\big] \approx \epsilon^2 \big\|J_{f_\theta}(x)\big\|_F^2. \tag{1}$$

Thus, minimizing $L_{cons}$ directly flattens the decision function in the vicinity of unsupervised samples. This illustrates clearly the mechanism by which consistency regularization encourages the network to move the decision boundary — and its surrounding region of high gradient — into regions of low sample density.

### 3.1 WHY SEMI-SUPERVISED SEMANTIC SEGMENTATION IS CHALLENGING

We attribute the infrequent success of consistency regularization in semantic segmentation problems to the observation that low density regions in input data do not align well with class boundaries. As illustrated in Figure 1, the cluster assumption is clearly violated: how much the raw pixel content of the receptive field of one pixel differ from the contents of the receptive field of a neighboring pixel has effectively no correlation with whether the patches' center pixels belong to the same class or not. For the cluster assumption to hold, we would require the distances between the contents of receptive fields to be large between classes and small within classes, which is not the case here.

The lack of variation in the patchwise distances is easy to explain from a signal processing perspective. With patch shape $B$, the $L^2$ distance between the pixel content of two overlapping patches centered at, say, horizontally neighboring pixels can be written as $\sqrt{B * (\Delta_x x)^2}$, where $*$ denotes convolution and $\Delta_x x$ is the horizontal gradient of the input image $x$. The squared gradient image is thus low-pass filtered by a $B$-shaped box filter, which suppresses any fine details and leads to smoothly varying sample density across the image. We test this experimentally in Appendix B.

Our decision to compare the raw pixel content of image patches is motivated by the fact that the perturbations used to generate $\hat{x}$ from $x$ in prior work operate in raw pixel (input) space. The success of virtual adversarial training (Miyato et al., 2017) in semi-supervised classification – in which $\hat{x}$ is

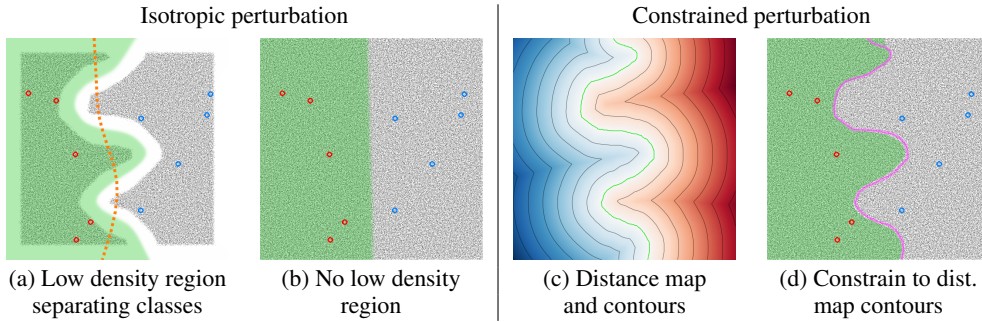

| Isotropic perturbation | | Constrained perturbation | |
|---|---|---|---|
| (a) Low density region separating classes | (b) No low density region | (c) Distance map and contours | (d) Constrain to dist. map contours |

Figure 2: Toy 2D semi-supervised classification experiments. Blue and red circles indicate supervised samples from class 0 and 1 respectively. The field of small black dots indicate unsupervised samples. The learned decision function is visualized by rendering the probability of class 1 in green; the soft gradation represents the gradual change in predicted class probability. (a, b) Semi-supervised learning with and without a low density region separating the classes. The dotted orange line in (a) shows the decision boundary obtained with plain supervised learning. (c) Rendering of the distance to the true class boundary with distance map contours. Strong colours indicate greater distance to class boundary. (d) Decision boundary learned when samples are perturbed along distance contours in (c). The magenta line indicates the true class boundary. Appendix A explains the setup in detail.

an adversarial example chosen to maximise the difference in prediction from that of $x$ – illustrates the significance of the input space distribution in semi-supervised learning.

## 3.2 CONSISTENCY REGULARIZATION WITHOUT CLUSTER ASSUMPTION

Let us analyze a simple toy example of learning to classify 2D points in a semi-supervised fashion. Figure 2a illustrates the setup where cluster assumption holds and there is a gap between the unsupervised samples belonging to the two different classes. The perturbation used for the consistency loss is a simple Gaussian nudge to both coordinates, and as expected, the learned decision boundary settles neatly between the two clusters.

In Figure 2b, the cluster assumption is violated and there are no density differences in the set of unsupervised samples. In this case, the consistency loss does more harm than good—even though it successfully flattens the neighborhood of the decision function, it does so also across the true class boundary. In order for the consistency regularization to be a net win, it would have to perturb the samples as much as possible, but at the same time avoid crossing the true class boundary.

In Figure 2c, we plot the contours of the distance to the true class boundary, suggesting a potentially better mechanism for perturbation. Indeed, when perturbations are done only along these contours, the probability of crossing the true class boundary is negligible compared to the regularization potential in the remaining dimension. Figure 2d shows that the resulting learned decision boundary aligns well with the true class boundary.

Low-density regions provide an effective signal that guides consistency regularization by providing areas into which a decision boundary can settle. This illustrative toy example demonstrates an alternative mechanism; the orientation of the decision boundary can be constrained to lie parallel to the directions of perturbation. We therefore argue that consistency regularization can be successful even when the cluster assumption is violated, if the following guidelines are observed: 1) the perturbations must be varied in order to cover as much of the input space in the same class as possible, 2) they must be high-dimensional in order to sufficiently constrain the orientation of the decision boundary in the high-dimensional space of natural imagery, 3) the probability of a perturbation crossing the true class boundary must be very small compared to the amount of exploration in other dimensions, and 4) the perturbed inputs should be plausible, i.e., they should not be grossly outside the manifold of real inputs.

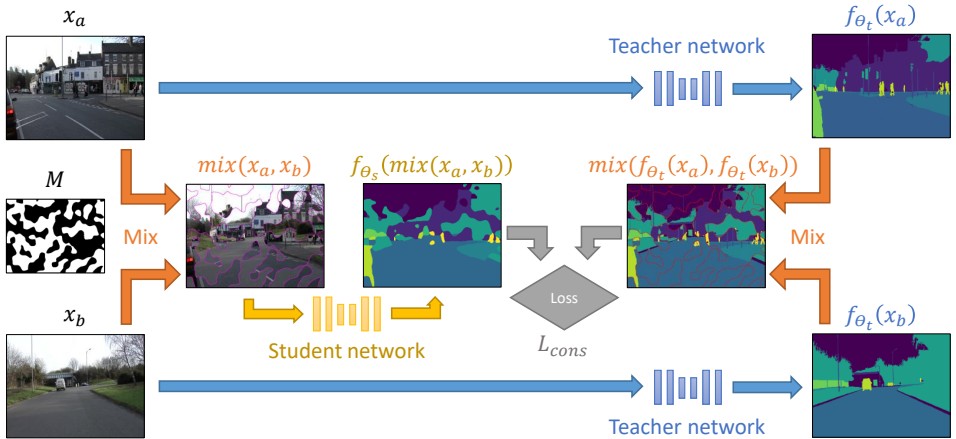

Figure 3: Illustration of mixing regularization for semi-supervised semantic segmentation with the mean teacher framework. $\theta_s$ and $\theta_t$ denote the weights of the student and teacher networks, respectively. The arbitrary mask $M$ is omitted from the argument list of function $mix$ for legibility.

### 3.3 CUTOUT AND CUTMIX FOR SEMANTIC SEGMENTATION

If we consider the classical augmentation-like perturbations such as translation, rotation, scaling, and brightness/contrast changes, it is evident that these have a low chance of confusing the output class (Athiwaratkun et al., 2019) but they also provide very little variation. Note that in context of semantic segmentation, all geometric transformations need to be applied in reverse for the result image before computing the loss (Ji et al., 2018). As such, translation turns into a no-op, unlike in classification tasks where it remains a useful perturbation (see Appendix D.1.1 for more details). Adding noise is another questionable perturbation strategy — although high-dimensional, such perturbations are very unlikely to lie on the manifold of natural images.

Out of previously proposed perturbation methods for consistency regularization, we identify CutOut and CutMix as promising candidates for semantic segmentation as they provide a large variety of possible outputs and are class preserving. Both approaches use a mask $M$ with a randomly chosen rectangular region. Our masks have $M = 1$ inside the rectangle, and $M = 0$ otherwise.

**CutOut.** To apply CutOut in a semantic segmentation task, we mask the input pixels with $M$ and disregard the consistency loss for pixels masked to 0 by $M$. Using square distance as the metric, we have $L_{cons} = \|M \odot (f_\theta(M \odot x) - f_\theta(x))\|^2$, where $\odot$ denotes an elementwise product.

**CutMix.** CutMix requires two input images that we shall denote $x_a$ and $x_b$ that we mix with the mask $M$. Following ICT (Verma et al. (2019)) we run both input images as well as the mixed variant through the network. The consistency loss is then taken between the segmentation of the mixed image and the mix between the segmented input images. To simplify the notation, let us define function $mix(a, b, M) = (1 - M) \odot a + M \odot b$ that selects the output pixel based on mask $M$. We can now write the consistency loss for segmentation CutMix as

$$L_{cons} = \left\|mix\big(f_\theta(x_a), f_\theta(x_b), M\big) - f_\theta\big(mix(x_a, x_b, M)\big)\right\|^2. \tag{2}$$

So far we have assumed that both original and perturbed images are segmented using the same network $f_\theta$. In this sense, our approach is similar to the $\Pi$-network Laine & Aila (2017), although mean teacher (Tarvainen & Valpola, 2017) is known to outperform it in image classification tasks. Our preliminary tests indicated that the same is true for semantic segmentation, and therefore all the experiments in this paper use the mean teacher framework. Specifically, in CutOut, we segment the unperturbed image $x$ using the teacher network and the perturbed image $M \odot x$ using the student network. In CutMix — following Verma et al. (2019) — we similarly segment the original images $x_a$ and $x_b$ using the teacher network, and the mixed image using the student network. The computation is illustrated in Figure 3.

To analyze how much variation these perturbations provide, we note that the original CutOut always masks out a fixed-size square at a random position, so the resulting mask has only two degrees

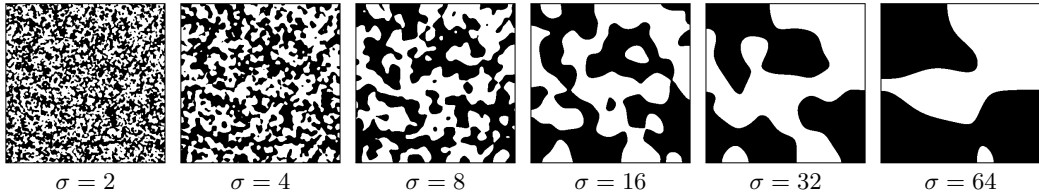

$$\sigma = 2 \qquad \sigma = 4 \qquad \sigma = 8 \qquad \sigma = 16 \qquad \sigma = 32 \qquad \sigma = 64$$

Figure 4: Example CowOut/CowMix masks with $p = 0.5$ and varying $\sigma$ in a 384×384 pixel image.

of freedom. The original CutMix has three degrees of freedom in choosing the position and size of the rectangle (the aspect ratio is fixed), and in addition it replaces the contents of the rectangle from another training image, providing one further degree of freedom. Probably more importantly, the CutOut-perturbed images are rather implausible because real data rarely contains axis-aligned constant-color rectangles, whereas the content-filled rectangles of CutMix are not nearly as conspicuous. In semantic segmentation, we deviate from the original methods in that we choose all four parameters of the rectangle at random in order to obtain as much variation as possible.

### 3.4 COWOUT AND COWMIX

The use of a rectangular mask restricts the dimensionality of the perturbations that CutOut and CutMix can produce. Intuitively, a more complex mask that has more degrees of freedom should provide better exploration of the plausible input space. We propose combining the semantic CutOut and CutMix regularizers introduced above with a novel mask generation method, giving rise to two regularization methods that we dub CowOut and CowMix due to the Friesian cow -like texture of the masks.

To generate a mask with a given proportion $p$ of pixels having $M = 0$, we start by sampling a Gaussian noise image $\mathcal{N}(0, I)$, convolve it with a Gaussian smoothing kernel, and threshold the result at $\tau = \mathrm{erf}^{-1}(2p - 1)\sqrt{2}S_\sigma + S_\mu$, where $S_\mu$ and $S_\sigma$ are the mean and standard deviation of the smoothed noise, respectively. The standard deviation $\sigma$ of the Gaussian smoothing kernel determines the average size of features in the mask and is drawn from a log-uniform distribution: $\sigma \sim \log\mathcal{U}(\sigma_{min}, \sigma_{max})$. Figure 4 shows example masks with varying values of $\sigma$.

To estimate how much variation these masks contain, we first observe that if no smoothing were performed before thresholding, we would obtain a binary per-pixel Bernoulli mask. This has an exponential amount of variation with respect to the area of the receptive field: a receptive field with area $A$ has $2^A$ possible masks. Applying the smoothing filter has a similar effect as zooming the mask — if the smoothing constant $\sigma$ is doubled, we can expect to obtain roughly 4× fewer "bits" of variation (see, e.g., Figure 4, $\sigma = 8$ vs. $\sigma = 16$). The suitable value for $\sigma$ is thus an empirical tradeoff between the amount of variation, i.e., the strength of the resulting perturbation, and plausibility of the masked/mixed image. If too little filtering is performed, we obtain a huge amount of variation but the perturbed images are unrealistic, and with too much filtering the images are very realistic but the amount of variation is small. As a geometric interpretation, we hypothesize that the more varied masks help the network to cope with occlusions better than the rectangular masks that can mimic occlusions only along horizontal and vertical edges.

Considering potential future work, we note that it should be possible to improve the results further with a more principled method for choosing $\sigma$ so that its distribution is automatically adapted to the training data.

### 4 EXPERIMENTS

We will now describe describe our experiments and main results. We will start by describing the training setup, followed by an investigation of various perturbation methods in the context of semi-supervised semantic segmentation, and conclude with a comparison against the state-of-the-art.

| Sup. baseline | Std. aug[a] | ICT | Cutout | CutMix[b] | CowOut | CowMix | Fully sup. |
|---|---|---|---|---|---|---|---|
| $48.66\%_{\pm 1.80}$ | $46.24\%_{\pm 2.21}$ | $48.29\%_{\pm 2.01}$ | $53.09\%_{\pm 2.56}$ | $50.95\%_{\pm 2.49}$ | $52.93\%_{\pm 1.35}$ | $\mathbf{55.06\%}_{\pm 1.74}$ | $64.19\%_{\pm 0.41}$ |

Table 1: Measurements and ablation on CAMVID test set. Our results are mean intersection-over-union (mIoU) presented as $mean \pm std.dev$ computed from 10 runs. Training was run for 300 epochs, except in (a) standard augmentation and (b) CutMix experiments where convergence failures were observed. For those experiments, the results were taken after (a) 50 and (b) 100 epochs.

## 4.1 TRAINING SETUP

We use two segmentation networks in our experiments: 1) U-Net (Ronneberger et al., 2015) with a ResNet-50 (He et al., 2016) based encoder that was pre-trained using ImageNet and a randomly initialized decoder (Appendix D.2), and 2) DeepLab v2 network (Chen et al., 2017) based on ResNet-101 and pre-trained for semantic segmentation using the COCO (Lin et al., 2014) dataset, as used by Hung et al. (2018).

We minimise the sum of the supervised cross-entropy loss and consistency loss $L_{cons}$ with weights of 1 and 10 respectively. We use the ean teacher algorithm (Tarvainen & Valpola, 2017), as detailed in Appendix D.3. We replace the sigmoidal ramp-up of the consistency regularization weight (Laine & Aila, 2017; Tarvainen & Valpola, 2017) with the average thresholded confidence of the teacher network (see Appendix D.3.4), which increases as the training progresses (French et al., 2018). Our implementation uses the PyTorch framework and will be made available.

## 4.2 COMPARISON OF PERTURBATION METHODS

The CAMVID (Brostow et al., 2008) training set consists of 367 images. We chose 10 subsets of 30 labeled images that were used for the supervised loss, while all training images were used to compute the consistency loss. The U-net setup is used in this test.

The results for different perturbation methods are given in Table 1. Perturbations based on standard data augmentation (flips, rotations, brightness, etc., see Appendix D.1) and Interpolation Consistency Training (ICT) resulted in no measurable improvement in the mean IoU score compared to the baseline of supervised training using only the labeled samples. The Cutout and CutMix experiments used masks with a single random rectangle, and led to clear improvements over the baseline.

For CowOut and CowMix, we generated the masks using $\sigma \in [4, 16]$. For CowOut, we found that choosing the masked pixel proportion randomly with $p \sim \mathcal{U}(0, 1)$ produced the best results, whereas for CowMix it was optimal to always use $p = 0.5$. CowMix led to the best result in this experiment, bridging approximately half of the gap between the supervised baseline and the fully supervised reference result.

We tuned all hyper-parameters using the CAMVID dataset due to its small size and fast run-time. We found that these hyper-parameters also worked well for other, larger datasets.

## 4.3 RESULTS ON CITYSCAPES AND PASCAL VOC

We will now compare our results against the state-of-the-art in semi-supervised semantic segmentation, which is currently the adversarial training approach of Hung et al. (2018). We use two datasets in our experiments. CITYSCAPES consists of urban scenery and has 2975 images in its training set. PASCAL VOC 2012 (Everingham et al., 2012) is more varied, but includes only 1464 training images, and thus we follow the lead of Hung et al. and augment it using SEMANTIC BOUNDARIES (Hariharan et al., 2011), resulting in 10582 training images. The practical benefits of semi-supervised learning are realised by reducing the required quantity of supervised training data as much as possible, hence we tested our approch using less supervised data than in Hung et al. (2018). We note that we generated CowMix masks using $\sigma \in [8, 32]$ due to the larger visual features present in CITYSCAPES and PASCAL VOC.

Our results are given in Tables 2 and 3 as mean intersection-over-union (mIoU) percentages, where higher is better. The supervised baseline results between Hung et al. and our DeepLab implementa-

| Labeled samples | 100 | 372 (12.5%) | 744 (25%) | 1488 (50%) | 2975 (All) |
|---|---|---|---|---|---|
| | Hung et al. (2018): Adversarial training with COCO-pretrained DeepLab v2 network | | | | |
| Baseline | — | 55.5% | 59.9% | 64.1% | 66.4% |
| Semi-supervised | — | 58.8% | 62.3% | 65.7% | — |
| - Delta | — | 3.5 | 2.4 | 1.6 | — |
| | Our results: Same COCO-pretrained DeepLab v2 network | | | | |
| Baseline | $44.34\%_{\pm 1.61}$ | $56.02\%_{\pm 0.80}$ | $60.90\%_{\pm 0.70}$ | $65.05\%_{\pm 0.71}$ | $67.79\%_{\pm 0.32}$ |
| ICT | | $55.30\%_{\pm 1.14}$ | | | |
| CutOut | | $54.25\%_{\pm 0.69}$ | | | |
| CutMix | | $57.64\%_{\pm 0.91}$ | | | |
| CowOut | | $58.09\%_{\pm 1.37}$ | | | |
| CowMix | $49.01\%_{\pm 2.58}$ | $60.53\%_{\pm 0.29}$ | $64.10\%_{\pm 0.82}$ | $66.51\%_{\pm 0.45}$ | $69.03\%_{\pm 0.27}$ |
| - Delta | 4.67 | 4.51 | 3.20 | 1.46 | 1.24 |
| | Our results: U-Net and randomly initialized decoder | | | | |
| Baseline | $43.83\%_{\pm 0.99}$ | $54.76\%_{\pm 0.50}$ | $60.35\%_{\pm 0.93}$ | $64.93\%_{\pm 0.32}$ | $68.34\%_{\pm 0.76}$ |
| CowMix | $\mathbf{51.98\%}_{\pm 2.76}$ | $\mathbf{61.48\%}_{\pm 1.84}$ | $\mathbf{64.85\%}_{\pm 0.26}$ | $\mathbf{66.90\%}_{\pm 0.22}$ | $\mathbf{69.57\%}_{\pm 0.43}$ |
| - Delta | 8.15 | 6.72 | 4.50 | 1.97 | 1.23 |

Table 2: Performance (mIoU) on CITYSCAPES validation set, each computed from 5 runs. The results for Hung et al. (2018) are from their paper.

| # Labels | 100 | 200 | 400 | 800 | 2646 (25%) | 10582 (All) |
|---|---|---|---|---|---|---|
| | Hung et al. (2018): Adversarial training with COCO-pretrained DeepLab v2 network | | | | | |
| Baseline | $39.22\%_{\pm 2.08}$ | $46.57\%_{\pm 1.34}$ | $55.65\%_{\pm 0.88}$ | $62.54\%_{\pm 0.45}$ | $68.41\%_{\pm 0.29}$ | $72.50\%_{\pm 0.27}$ |
| Semi-sup. | $38.82\%_{\pm 3.91}$ | $49.40\%_{\pm 1.00}$ | $60.29\%_{\pm 2.25}$ | $66.45\%_{\pm 0.69}$ | $\mathbf{71.27\%}_{\pm 0.26}$ | — |
| - Delta | 0.40 | 2.83 | 4.64 | 3.91 | 2.86 | — |
| | Our results: Same COCO-pretrained DeepLab v2 network | | | | | |
| Baseline | $41.17\%_{\pm 1.96}$ | $48.96\%_{\pm 1.70}$ | $58.18\%_{\pm 1.16}$ | $64.51\%_{\pm 0.85}$ | $70.16\%_{\pm 0.35}$ | $73.32\%_{\pm 0.19}$ |
| CowMix | $\mathbf{52.10\%}_{\pm 1.35}$ | $\mathbf{57.84\%}_{\pm 1.55}$ | $\mathbf{64.18\%}_{\pm 1.79}$ | $\mathbf{67.84\%}_{\pm 0.88}$ | $70.99\%_{\pm 0.41}$ | $\mathbf{73.43\%}_{\pm 0.06}$ |
| - Delta | 10.93 | 8.88 | 6.00 | 3.33 | 0.83 | 0.11 |
| | Our results: U-Net and randomly initialized decoder | | | | | |
| Baseline | $24.91\%_{\pm 2.13}$ | $33.66\%_{\pm 1.64}$ | $41.80\%_{\pm 1.15}$ | $49.18\%_{\pm 1.61}$ | $60.58\%_{\pm 0.77}$ | $65.97\%_{\pm 1.11}$ |
| CowMix | $41.00\%_{\pm 2.70}$ | $49.42\%_{\pm 1.74}$ | $54.20\%_{\pm 1.12}$ | $54.67\%_{\pm 5.66}$ | $63.48\%_{\pm 0.26}$ | $65.54\%_{\pm 4.99}$ |
| - Delta | 16.09 | 15.76 | 12.4 | 5.49 | 2.90 | -0.43 |

Table 3: Performance (mIoU) on augmented PASCAL VOC validation set, each computed from 5 runs.

tion are based on the same setup, provided by the authors, but differ slightly in practice due to the different choice of optimizer, etc.

We can see that using the same DeepLab setup, CowMix outperforms the adversarial training approach in both datasets, with the exception of PASCAL when using a large number of labeled samples. The difference is particularly significant when only a small number of labeled samples is available, e.g., 52% vs 38% mIoU with 100 labeled samples in PASCAL.

When using our U-Net architecture — in which the decoder *has not been pre-trained* — we were unable to successfully apply the adversarial approach. U-Net improves the performance of CowMix on CITYSCAPES and continues to yield significant gains relative to the supervised baseline in the case of PASCAL. While the performance on the more varied PASCAL dataset significantly benefits the pre-training of DeepLab v2, it also provides a good test environment for semi-supervised learning algorithms when the decoder is randomly initialized.

## 4.4 DISCUSSION

We attribute the fact that the adversarial approach (Hung et al. (2018)) is ineffective with small numbers of labeled samples to the requirements imposed by its discriminator network. A small set of ground truth labels lacks the variation necessary to effectively train the discriminator to distinguish

ground truth from predicted segmentation maps, preventing it from effectively guiding the segmentation network. In contrast, consistency regularization minimizes variation in prediction over class preserving perturbation, effectively propagating labels between unlabelled samples. It therefore does not impose similar requirements on the size of the labeled data set.

## 5 CONCLUSIONS

We have shown that consistency regularization is a viable solution for semi-supervised semantic segmentation, despite the lack of low-density regions between classes. The proposed CowMix regularization leads to very high-dimensional perturbations that enable state-of-the-art results, while being considerably easier to implement and use than the previous methods based on adversarial training. Even better and more varied perturbation strategies are an obvious avenue for future work. Additionally, it could be fruitful to investigate when the CowMix approach could also be beneficial in the context of classification.

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

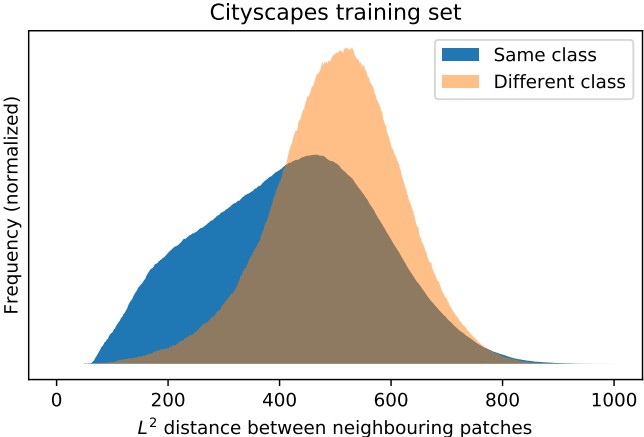

Figure 5: Histogram of $L^2$ distance between pairs of patches of size $225 \times 225$ centred on neighboring pixels, split between those with the same class in blue and pairs that lie either side of a class boundary in orange. Computed from all patches in the CITYSCAPES dataset whose central pixel has a valid ground truth class.

## A    2D TOY EXPERIMENTS

The neural networks used in our 2D toy experiments are simple classifiers in which samples are 2D $x, y$ points ranging from -1 to 1. Our networks are multi-layer perceptrons consisting of 3 hidden layers of 512 units, each followed by a ReLU non-linearity. The final layer is a 2-unit classification layer. We use the mean teacher (Tarvainen & Valpola, 2017) semi-supervised learning algorithm with binary cross-entropy as the consistency loss function, a consistency loss weight of 10 and confidence thresholding (French et al., 2018) with a threshold of 0.97.

The ground truth decision boundary was derived from a hand-drawn $512 \times 512$ pixel image. The distance map shown in Figure 2(c) was computed using `scipy.ndimage.morphology.distance_transform_edt`, with distances negated for regions assigned to class 0. Each pixel in the distance map therefore has a signed distance to the ground truth class boundary. This distance map was used to generate the countours seen as lines in Figure 2(c) and used to support the constrained consistency regularization experiment illustrated in Figure 2(d).

The constrained consistency regularization experiment described in Section 3.2 required that a sample $x$ should be perturbed to $\hat{x}$ such that they are at the same — or similar — distance to the ground truth decision boundary. This was achieved by drawing isotropic perturbations from a normal distrubtion $\hat{x} = x + h$ where $h \sim \mathcal{N}(0, 0.117)$ ($0.117 \approx 30$ pixels in the source image), determining the distances $m(x)$ and $m(\hat{x})$ from $x$ and $\hat{x}$ to the ground truth boundary (using a precomputed distance map) and discarding the perturbation – by masking consistency loss for $x$ to 0 – if $|m(\hat{x}) - m(x)| > 0.016$ ($0.016 \approx 4$ pixels in the source image).

## B    DISTRIBUTION OF PATCHES IN THE CITYSCAPES DATASET

In Section 3.1 we observed that if we extract overlapping neighboring patches from an image, where the patches are centered on neighboring pixels and compute the $L^2$ distance between the raw pixel content of neighboring patches, we see that the distances vary smoothly across the image. This is due to the fact that the distance between these patches can be efficiently computed by applying a box/uniform filter to the squared gradient image. This equivalence between the patch-to-patch distance and a uniform filter (that suppresses high frequencies and therefore the fine details necessary for low-density regions / large inter-patch distances to lie along class/object/texture boundaries in an image) suggests that these distance maps will be smooth for natural images in general.

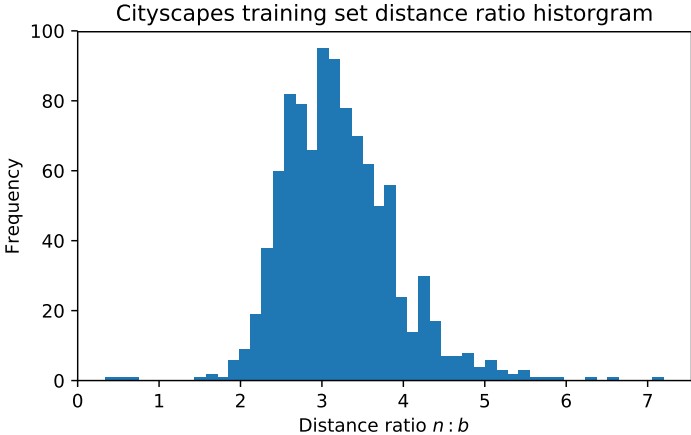

Figure 6: Histogram of the ratio of the distances $\frac{n}{b}$, where $n$ is the $L^2$ pixel content distance from the query patch $Q_i$ drawn from image $I_j$ to the nearest neighbour / most similar patch drawn from image $I_k$ such that $k \neq j$ and $b$ is the distance from the query patch $Q_i$ to the patch $P_i$ centred on a neighbouring pixel that has a different ground truth class. Put simply, this histogram illustrates that the closest patch to $Q_i$ found in any other image in CITYSCAPES is typically at around $3\times$ the distance as an immediate neighbour that lies on the other side of a class boundary.

We will now use the CITYSCAPES dataset to experimentally test our observations concerning the distribution of image patches. Figure 5 illustrates the distributions of distances between pairs of patches centred on neighbouring pixels that have the same class in comparison to pairs of patches centred on pixels that lie either side of a class boundary. The higher proportion of lower pixel content distances between neighboring patches that have the same class can be explained by the presence of large lightly textured regions such as sky that have few class boundaries and patches with similar pixel content.

Further analysis of the CITYSCAPES dataset revealed that in terms of pixel content distance, the nearest neighbours to a patch are other patches centred on neighboring and nearby pixels. We chose 1000 pairs of patches $Q_i$ and $P_i$ where $i \leq 1000$ from CITYSCAPES whose central pixels lie either side of a class boundary. For each query patch $Q_i$ we computed the distance to all other patches in the entire dataset. Figure 6 shows that the closest neighbour extracted from another image is typically around $3\times$ the distance from $Q_i$ as the immediate neighbor $P_i$.

### B.1 EFFICIENT COMPUTATION OF SLIDING WINDOW PATCH DISTANCES

Computing the distance from patch $Q_i$ to every patch centred on each pixel in the complete CITYSCAPES dataset would be prohibitively expensive if performed in a brute-force fashion. There are 2975 training images in CITYSCAPESwith each image having a resolution of $1024 \times 512$ pixels (downsampled by a factor of 2 from the original resolution, as used in our segmentation experiments). Centering a patch on each pixel results in 1,559,756,800 patches.

Given a query patch $Q$ and a patch $A$ where $A$ is extracted from an image $I$, the squared distance $\|A - Q\|^2$ can be expanded to $Q \cdot Q + A \cdot A - 2(Q \cdot A)$. $Q \cdot Q$ is a constant, $A \cdot A$ can be computed for all patches in $I$ in sliding window fashion using the box filter trick from Section 3.1; $A \cdot A = B * I^2$ where $B$ is a box-filter kernel, $I^2$ is the element-wide squared image and $*$ represents convolution. $Q \cdot A$ can be computed using correlation; given that the correlation kernel $Q$ is typically large, we use FFT-based convolution.

## C PSEUDOCODE

The mask generation function is given in the form of Python/PyTorch code in Listing 1, and the training functions for semi-supervised CutOut and CutMix are given in Listings 2 and 3, respectively.

The latter incorporate the mean teacher framework (Tarvainen & Valpola, 2017) and confidence thresholding (French et al., 2018) as used in our experiments. In keeping with Yun et al. (2019); DeVries & Taylor (2017); Verma et al. (2019), our mask generator allows the proportion of pixels that come from each source image to vary. Although we found via experimentation that for CowMix the best proportion was always $p = 0.5$, varying the proportion is beneficial for CowOut as detailed in Section 4.2.

```python
import numpy as np
from scipy.ndimage.filter import gaussian_filter
from scipy.special import erfinv

def generate_mixing_mask(img_size, sigma_min, sigma_max, p_min, p_max):
    # Randomly draw sigma from log-uniform distribution
    sigma = np.exp(np.random.uniform(np.log(sigma_min), np.log(sigma_max)))
    p = np.random.uniform(p_min, p_max)        # Randomly draw proportion p
    N = np.random.normal(size=img_size)        # Generate noise image
    Ns = gaussian_filter(N, sigma)             # Smooth with a Gaussian
    # Compute threshold
    t = erfinv(p*2 - 1) * (2**0.5) * Ns.std() + Ns.mean()
    return (noise_smooth > t).astype(float)  # Apply threshold and return
```
Listing 1: Python/NumPy code for cow mask generation

```python
def cutout_loss(x, mask, teacher_model, student_model):
    """x is input image of shape (batch, chan, H, W), mask is mixing mask
       of shape (batch, 1, H, W)"""
    # Apply teacher model and softmax to get per-pixel class probability
    y_t = softmax(teacher_model(x), dim=1)
    # Apply student model to masked image
    y_s = softmax(student_model(x * mask), dim=1)
    # Confidence thresholding factor
    confidence = y_t.max(dim=1) # Dimension 1 is class prob.
    conf_fac = (confidence > 0.97).mean()
    # Consistency is squared error between student and teacher preds for
    #   masked pixels only, modulated with confidence factor
    return (squared_diff(ym_t, ym_s) * m).mean() * conf_fac
```
Listing 2: PyTorch code for CutOut / CowOut for segmentation

```python
def cutmix_loss(xa, xb, mask, teacher_model, student_model):
    """xa, xb are input image pair each of shape (batch, chan, H, W), mask
       is mixing mask of shape (batch, 1, H, W)"""
    # Apply teacher model and softmax to get per-pixel class probability
    ya_t = softmax(teacher_model(xa), dim=1)
    yb_t = softmax(teacher_model(yb), dim=1)
    # Mix images and teacher predictions
    xm = xa * (1 - mask) + xb * mask
    ym_t = ya_t * (1 - mask) + yb_t * mask
    # Apply student model to mixed image
    ym_s = softmax(student_model(xm), dim=1)
    # Confidence thresholding factor
    confidence = ym_t.max(dim=1) # Dimension 1 is class prob.
    conf_fac = (confidence > 0.97).mean()
    # Consistency is squared error between student and teacher preds,
    #   modulated with confidence factor
    return squared_diff(ym_t, ym_s).mean() * conf_fac
```
Listing 3: PyTorch code for CutMix / CowMix for segmentation

## D  SEMANTIC SEGMENTATION EXPERIMENTS

### D.1  DATA AUGMENTATION

Our data augmentation scheme that we used for standard augmentation based perturbation consists of an affine transformation composed of horizontal flips, translation in the range $[-4, 4]$ pixels,

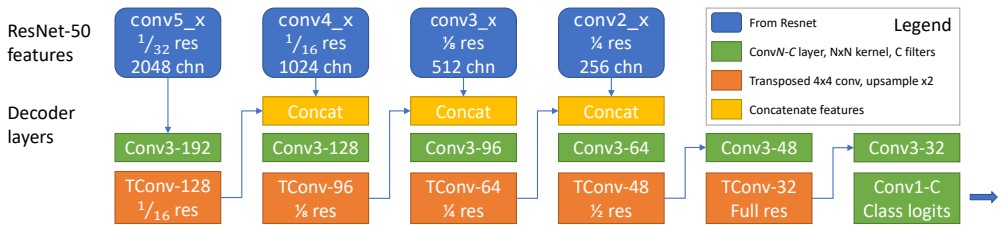

Figure 7: ResNet-50 based U-Net decoder architecture.

uniform scaling in the range $[0.8, 1.25]$ and rotation in the range $[-15°, 15°]$. We also modify the brightness and contrast by adding a value $b \sim \mathcal{N}(0, 0.1)$ and scaling by a factor $c \sim e^{\mathcal{N}(0, ln(1.1))}$.

### D.1.1 AFFINE TRANSFORMATION BASED DATA AUGMENTATION FOR SEMANTIC SEGMENTATION

In Section 3.3 we stated that for the purpose of semantic segmentation, all geometric transformations need to be applied in reverse for the result image before computing the loss and that translation effectively becomes a no-op. This occurs in a similar manner in both the supervised and unsupervised pathways.

Let us assume that the image $x$ is augmented by applyng the affine transformation matrix $M$; $\hat{x} =$ transform$(x, A)$. We apply our segmentation network $f_\theta$ to get our predictions $\hat{y}_{pred} = f_\theta(\hat{x})$. We note that the pixel-wise predictions $\hat{y}_{pred}$ were generated from the transformed image $\hat{x}$ and are therefore are not aligned with our ground truth annotations $y$. There are two ways in which we can compute the supervised loss $L_{ce}$; we can apply the transformation $A$ to the ground truths $\hat{y} =$ transform$(y, A)$ to align them with $\hat{y}_{pred}$ and compute $L_{ce} =$ cross_entropy$(\hat{y}_{pred}, y)$ or we can apply the inverse transformation $A^{-1}$ to our predictions to bring them into alignment with the original image $y_{pred} =$ transform$(\hat{y}_{pred}, A^{-1})$. Should the transformation $A$ include a translation, it will be countered by applying the equivalent translation to the pixel-wise ground truth in the first case or will be undone by applying $A^{-1}$ to the predictions in the second case.

We must adopt a similar strategy when computing the unsupervised loss term $L_{cons}$ when using affine augmentation as perturbation. If the image $x$ is transformed using two different affine transformations $A$ and $B$ producing $\hat{x}$ and $\tilde{x}$. Our network $f_\theta$ will generate predictions $\hat{y}_{pred}$ and $\tilde{y}_{pred}$ that are not aligned with one another. In order to compute the consistency loss $L_{cons}$ we must align the pixel-wise predictions with one another. This can be done by transforming $\tilde{y}_{pred}$ into the same frame of reference as $\hat{y}_{pred}$ by applying the transformation $AB^{-1}$; $\bar{y}_{pred} =$ transform$(\tilde{y}_{pred}, AB^{-1})$.

We would like to note that these constraints limit the utility of translations used as an augmentation for semantic segmentation. Classification problems are typically translation invariant; applying a translation will not change the target class. This is not true for semantic segmentation which is translation variant. For example, consider applying a translation to a patch that is centered on a pixel that is very close to a class boundary that separates a car from a road in an image. The translated patch may now have a different class. Computing either supervised or unsupervised loss without countering this translation will either cause the network to learn from erroneous training data.

### D.2 U-NET NETWORK ARCHITECTURE

Our U-Net network is shown in Figure 7 and Table 4.

| Description | Resolution $\times$ channels |
|---|---|
| ResNet-50 layer `conv5_x` $\frac{1}{32}$ res, 2048 chn | $\frac{1}{32} \times 2048$ |
| Conv $3 \times 3 \times 192$ | $\frac{1}{32} \times 192$ |
| TransposeConv $4 \times 4 \times 128$ | $\frac{1}{16} \times 128$ |
| Concat with ResNet-50 layer `conv4_x` $\frac{1}{16}$ res, 1024 chn | $\frac{1}{16} \times 1152$ |
| Conv $3 \times 3 \times 128$ | $\frac{1}{16} \times 128$ |
| TransposeConv $4 \times 4 \times 96$ | $\frac{1}{8} \times 96$ |
| Concat with ResNet-50 layer `conv3_x` $\frac{1}{8}$ res, 512 chn | $\frac{1}{8} \times 608$ |
| Conv $3 \times 3 \times 96$ | $\frac{1}{8} \times 96$ |
| TransposeConv $4 \times 4 \times 64$ | $\frac{1}{4} \times 64$ |
| Concat with ResNet-50 layer `conv2_x` $\frac{1}{4}$ res, 256 chn | $\frac{1}{4} \times 320$ |
| Conv $3 \times 3 \times 64$ | $\frac{1}{4} \times 64$ |
| TransposeConv $4 \times 4 \times 48$ | $\frac{1}{2} \times 48$ |
| Conv $3 \times 3 \times 48$ | $\frac{1}{2} \times 48$ |
| TransposeConv $4 \times 4 \times 32$ | $1 \times 32$ |
| Conv $3 \times 3 \times 32$ | $1 \times 32$ |
| Conv $1 \times 1 \times C$ | $1 \times C$ |

Table 4: ResNet-50 based U-Net decoder. $C$ is the number of target classes.

## D.3 TRAINING DETAILS

### D.3.1 EXPERIMENTS USING U-NET ARCHITECTURE

In keeping with Long et al. (2015) we use a batch size of 1. We freeze the batch normalization layers within the ResNet encoder, using the pre-trained running mean and variance rather than computing per-batch mean and variance during training. We use the Adam Kingma & Ba (2015) optimization algorithm with a learning rate of $1 \times 10^{-4}$. As per the mean teacher algorithm Tarvainen & Valpola (2017), after each iteration the weights $w_t$ of the teacher network are updated to be the exponential moving average of the weights $w_s$ of the student: $w_t = \alpha_t w_t + (1 - \alpha_t)w_s$, where $\alpha_t = 0.99$.

The Cityscapes images were downsampled to half resolution ($1024 \times 512$) prior to use, as in Hung et al. (2018). When using Cityscapes we trained for 100,000 iterations using a batch size of 1.

We tuned our approach and selected hyper-parameters using the CAMVID dataset due to its small size and fast run-time, after which we applied the same hyper-parameters to Cityscapes. We did not run the augmentation based perturbation experiments on the Cityscapes dataset due to the long run-time involved.

### D.3.2 EXPERIMENTS USING DEEPLAB V2 ARCHITECTURE

We found that we had to decrease the learning rate to $1 \times 10^{-5}$ to get good performance with DeepLab v2. We note that the PyTorch implementation of DeepLab v3 worked best with $1 \times 10^{-4}$. We believe that this is because the U-Net and DeepLab v3 networks accept input images with zero-mean unit-variance, where DeepLab v2 accepts images whose values are in the range 0 to 255 with only mean subtraction.

For the PASCAL VOC experiments, we extracted $321 \times 321$ random crops and used a batch size of 10, in keeping with Hung et al. (2018). For the CITYSCAPES experiments we used full image crops and a batch size of 2.

### D.3.3 ABLATION

The augmentation based perturbation experiments performed on the CAMVID dataset were trained for 50 epochs. The CutMix and CowOut experiments were trained for 100.

### D.3.4 CONFIDENCE THRESHOLDING

French et al. (2018) apply confidence thresholding, in which they mask consistency loss to 0 for samples whose confidence as predicted by the teacher network is below a threshold of 0.968. In the

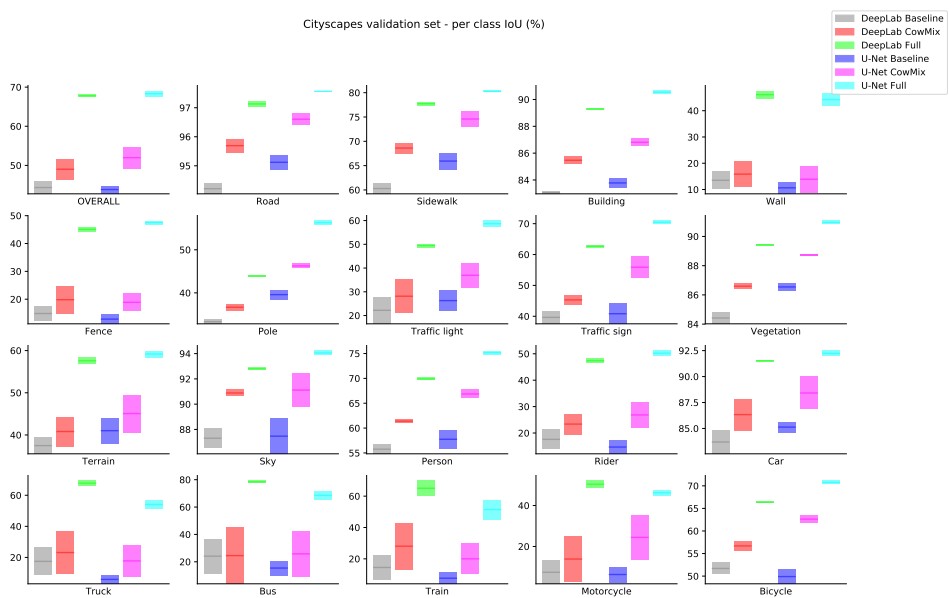

Figure 8: Visualization of semi-supervised segmentation on CITYSCAPES, 100 supervised samples

context of segmentation, we found that this masks pixels close to class boundaries as they usually have a low confidence. These regions are often large enough to encompass small objects, preventing learning and degrading performance. Instead we modulate the consistency loss with the proportion of pixels whose confidence is above the threshold. This values grows throughout training, taking the place of the sigmoidal ramp-up used in Laine & Aila (2017); Tarvainen & Valpola (2017).

### D.4  DETAILED PERFORMANCE TABLES

The detailed per-class performance on the CITYSCAPES dataset are presented in Table 5 and visualized in Figure 8, ad on the augmented PASCAL VOC dataset are presented in Table 6 and visualized in Figure 9.

### D.5  HYPER-PARAMETER CHOICE

We explored the hyper-parameters for CowMix using the CAMVID dataset. We present our results in the form of bar plots. Black and green bars at left and right end are the performance of the baseline and fully supervised setups respectively while blue bars show response to hyper-parameter values. The performance is strong when consistency weight has a value in the range of 3 to 30, with 10 being the optimum choice, as seen in Figure 10.

The EMA $\alpha$ value used to update the teacher network had little effect, as seen in Figure 11.

The $\sigma$ used for Gaussian smoothing yielded good results when between 4 and 8, but was best when drawn from $\sigma \sim \log \mathcal{U}(4, 16)$, as seen in Figure 12.

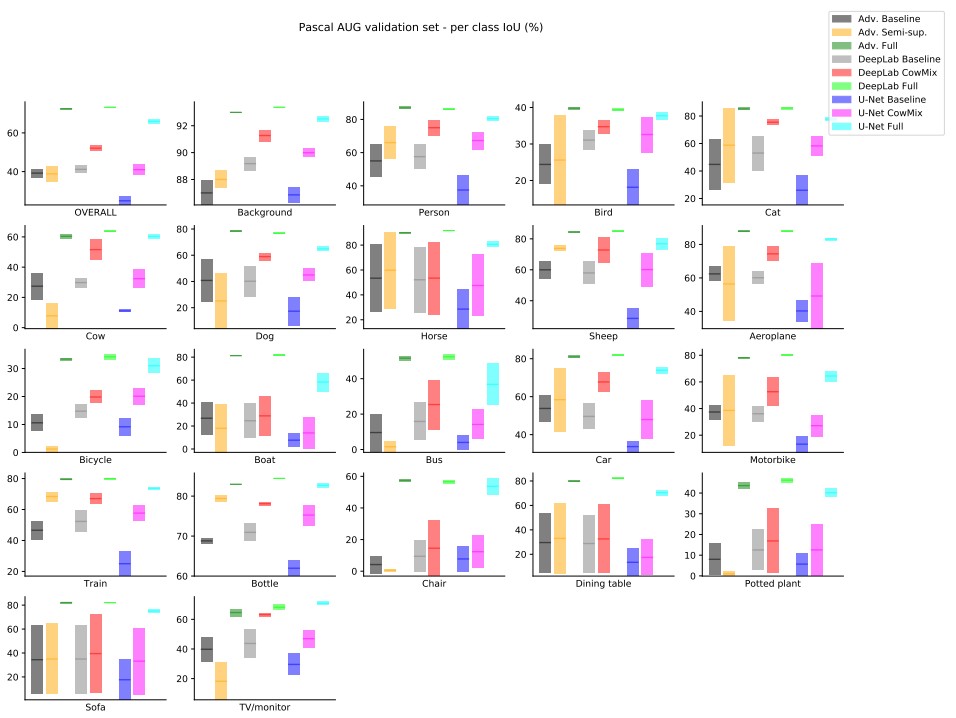

Figure 9: Visualization of semi-supervised segmentation on augmented PASCAL VOC dataset, 100 supervised samples

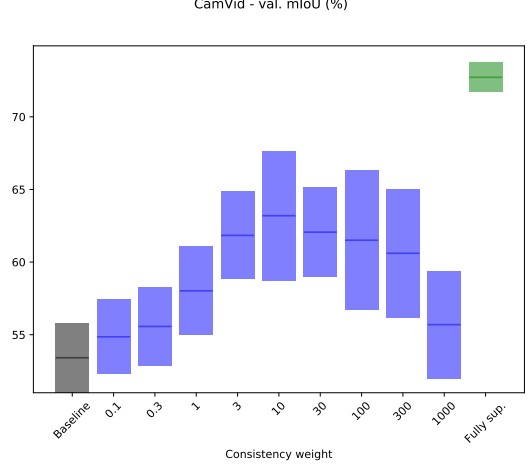

Figure 10: Effect of consistency weight hyper-parameter. In each bar, the central line is the mean, with the extents of the bar placed 1 standard deviation each side.

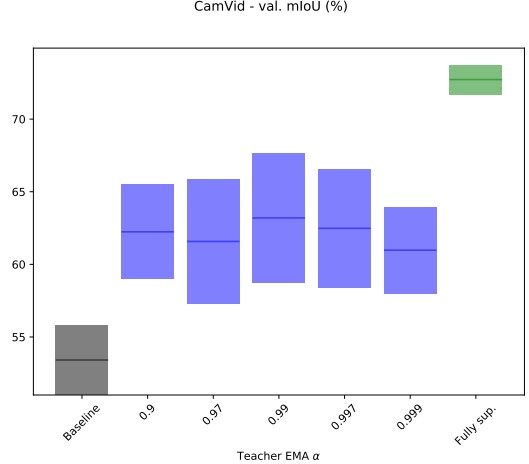

Figure 11: Effect of mean teacher EMA hyper-parameter.

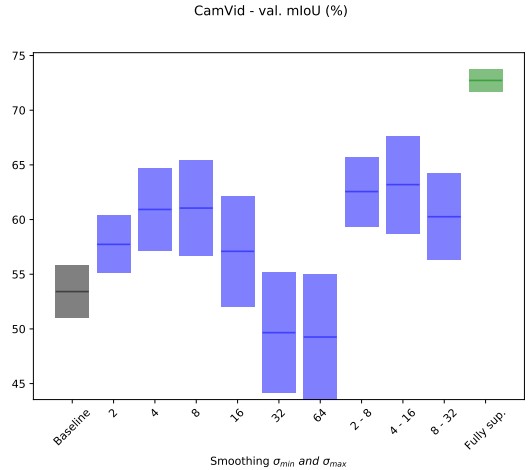

Figure 12: Effect of CowMix $\sigma_{min}$ and $\sigma_{max}$ hyper-parameters. A single value indicates that $\sigma$ was fixed, while a range (e..g 4-16) indicates $\sigma \sim \log \mathcal{U}(4, 16)$.

| | OVERALL | Road | Sidewalk | Building |
|---|---|---|---|---|
| DeepLab Baseline | $44.34\% \pm 1.61$ | $94.21\% \pm 0.18$ | $60.29\% \pm 1.09$ | $83.06\% \pm 0.09$ |
| DeepLab CowMix | $49.01\% \pm 2.58$ | $95.69\% \pm 0.24$ | $68.62\% \pm 1.11$ | $85.47\% \pm 0.30$ |
| DeepLab Full | $67.79\% \pm 0.32$ | $97.13\% \pm 0.09$ | $77.74\% \pm 0.36$ | $89.29\% \pm 0.07$ |
| U-net Baseline | $43.83\% \pm 0.99$ | $95.12\% \pm 0.25$ | $65.94\% \pm 1.74$ | $83.78\% \pm 0.34$ |
| U-net CowMix | $51.98\% \pm 2.76$ | $96.61\% \pm 0.20$ | $74.59\% \pm 1.57$ | $86.81\% \pm 0.27$ |
| U-net Full | $68.34\% \pm 0.76$ | $97.57\% \pm 0.02$ | $80.26\% \pm 0.16$ | $90.54\% \pm 0.12$ |
| | Wall | Fence | Pole | Traffic light |
| DeepLab Baseline | $13.51\% \pm 3.39$ | $14.86\% \pm 2.63$ | $33.31\% \pm 0.49$ | $22.20\% \pm 5.68$ |
| DeepLab CowMix | $15.85\% \pm 4.84$ | $19.82\% \pm 4.89$ | $36.66\% \pm 0.79$ | $28.14\% \pm 7.10$ |
| DeepLab Full | $46.12\% \pm 1.51$ | $45.09\% \pm 0.88$ | $43.91\% \pm 0.14$ | $49.49\% \pm 0.73$ |
| U-net Baseline | $10.65\% \pm 2.28$ | $12.84\% \pm 1.67$ | $39.59\% \pm 1.09$ | $26.29\% \pm 4.40$ |
| U-net CowMix | $13.91\% \pm 5.01$ | $18.87\% \pm 3.07$ | $46.25\% \pm 0.50$ | $36.94\% \pm 5.18$ |
| U-net Full | $44.22\% \pm 2.36$ | $47.52\% \pm 0.69$ | $56.29\% \pm 0.47$ | $58.68\% \pm 1.28$ |
| | Traffic sign | Vegetation | Terrain | Sky |
| DeepLab Baseline | $39.65\% \pm 2.08$ | $84.42\% \pm 0.40$ | $37.48\% \pm 1.94$ | $87.30\% \pm 0.75$ |
| DeepLab CowMix | $45.30\% \pm 1.48$ | $86.60\% \pm 0.20$ | $40.82\% \pm 3.46$ | $90.88\% \pm 0.24$ |
| DeepLab Full | $62.67\% \pm 0.39$ | $89.42\% \pm 0.04$ | $57.59\% \pm 0.79$ | $92.80\% \pm 0.08$ |
| U-net Baseline | $40.83\% \pm 3.27$ | $86.54\% \pm 0.27$ | $41.02\% \pm 3.02$ | $87.47\% \pm 1.40$ |
| U-net CowMix | $55.89\% \pm 3.44$ | $88.73\% \pm 0.05$ | $45.08\% \pm 4.41$ | $91.12\% \pm 1.31$ |
| U-net Full | $70.43\% \pm 0.55$ | $90.96\% \pm 0.12$ | $59.19\% \pm 0.80$ | $94.05\% \pm 0.18$ |
| | Person | Rider | Car | Truck |
| DeepLab Baseline | $55.76\% \pm 0.93$ | $17.59\% \pm 3.70$ | $83.70\% \pm 1.14$ | $17.46\% \pm 8.96$ |
| DeepLab CowMix | $61.39\% \pm 0.39$ | $23.38\% \pm 3.80$ | $86.33\% \pm 1.50$ | $23.15\% \pm 13.60$ |
| DeepLab Full | $69.89\% \pm 0.24$ | $47.37\% \pm 0.76$ | $91.52\% \pm 0.08$ | $67.77\% \pm 1.65$ |
| U-net Baseline | $57.75\% \pm 1.81$ | $14.69\% \pm 2.56$ | $85.13\% \pm 0.49$ | $5.86\% \pm 2.83$ |
| U-net CowMix | $66.86\% \pm 0.90$ | $26.85\% \pm 4.78$ | $88.42\% \pm 1.57$ | $17.77\% \pm 10.13$ |
| U-net Full | $75.18\% \pm 0.40$ | $50.23\% \pm 0.96$ | $92.23\% \pm 0.27$ | $54.06\% \pm 2.89$ |
| | Bus | Train | Motorcycle | Bicycle |
| DeepLab Baseline | $24.11\% \pm 12.62$ | $14.49\% \pm 7.71$ | $7.40\% \pm 5.63$ | $51.69\% \pm 1.24$ |
| DeepLab CowMix | $24.52\% \pm 20.60$ | $28.07\% \pm 14.68$ | $13.86\% \pm 11.17$ | $56.69\% \pm 1.10$ |
| DeepLab Full | $78.46\% \pm 0.86$ | $65.02\% \pm 4.88$ | $50.41\% \pm 1.75$ | $66.36\% \pm 0.13$ |
| U-net Baseline | $15.42\% \pm 5.23$ | $7.65\% \pm 3.57$ | $6.31\% \pm 3.76$ | $49.90\% \pm 1.60$ |
| U-net CowMix | $25.80\% \pm 16.62$ | $20.02\% \pm 9.65$ | $24.43\% \pm 10.97$ | $62.63\% \pm 0.88$ |
| U-net Full | $68.56\% \pm 3.08$ | $51.44\% \pm 6.32$ | $46.26\% \pm 1.31$ | $70.74\% \pm 0.41$ |

Table 5: Per-class performance on CITYSCAPES dataset, 100 supervised samples

| | OVERALL | Background | Person | Bird |
|---|---|---|---|---|
| Adv. Baseline | 39.22% ± 2.08 | 87.00% ± 0.89 | 55.00% ± 9.88 | 24.40% ± 5.46 |
| Adv. Semi-sup. | 38.82% ± 3.91 | 88.00% ± 0.63 | 66.00% ± 9.70 | 25.60% ± 12.22 |
| Adv. Full | 72.50% ± 0.27 | 93.00% ± 0.00 | 87.00% ± 0.63 | 39.80% ± 0.40 |
| DeepLab Baseline | 41.17% ± 1.96 | 89.18% ± 0.51 | 57.57% ± 7.40 | 31.07% ± 2.66 |
| DeepLab CowMix | 52.10% ± 1.35 | 91.27% ± 0.43 | 75.01% ± 4.77 | 34.73% ± 1.90 |
| DeepLab Full | 73.32% ± 0.19 | 93.39% ± 0.03 | 86.13% ± 0.39 | 39.36% ± 0.39 |
| U-net Baseline | 24.91% ± 2.13 | 86.85% ± 0.57 | 37.48% ± 9.01 | 18.15% ± 4.85 |
| U-net CowMix | 41.00% ± 2.70 | 89.99% ± 0.32 | 67.28% ± 5.26 | 32.59% ± 4.92 |
| U-net Full | 65.97% ± 1.11 | 92.50% ± 0.19 | 80.44% ± 1.45 | 37.74% ± 1.01 |

| | Cat | Cow | Dog | Horse |
|---|---|---|---|---|
| Adv. Baseline | 44.80% ± 18.54 | 27.40% ± 8.64 | 40.80% ± 16.27 | 53.40% ± 27.22 |
| Adv. Semi-sup. | 58.80% ± 26.95 | 7.80% ± 8.47 | 25.20% ± 21.17 | 59.80% ± 30.72 |
| Adv. Full | 85.20% ± 0.98 | 60.40% ± 1.62 | 78.40% ± 0.49 | 90.00% ± 0.63 |
| DeepLab Baseline | 53.06% ± 12.50 | 29.79% ± 3.06 | 40.19% ± 11.65 | 52.14% ± 26.38 |
| DeepLab CowMix | 75.49% ± 2.11 | 51.59% ± 6.62 | 58.94% ± 2.76 | 53.47% ± 29.03 |
| DeepLab Full | 85.84% ± 0.91 | 63.73% ± 0.52 | 76.76% ± 0.44 | 91.58% ± 0.17 |
| U-net Baseline | 26.02% ± 10.70 | 11.06% ± 0.69 | 17.34% ± 10.69 | 28.56% ± 15.70 |
| U-net CowMix | 58.26% ± 7.16 | 32.37% ± 6.23 | 44.96% ± 4.58 | 47.54% ± 24.69 |
| U-net Full | 77.76% ± 1.16 | 60.25% ± 1.48 | 64.85% ± 1.60 | 80.54% ± 2.36 |

| | Sheep | Aeroplane | Bicycle | Boat |
|---|---|---|---|---|
| Adv. Baseline | 60.00% ± 5.76 | 62.40% ± 4.45 | 10.60% ± 3.01 | 26.80% ± 14.23 |
| Adv. Semi-sup. | 73.80% ± 1.72 | 56.40% ± 22.29 | 1.20% ± 1.17 | 18.00% ± 20.94 |
| Adv. Full | 84.40% ± 0.49 | 87.80% ± 0.40 | 33.40% ± 0.49 | 81.20% ± 0.40 |
| DeepLab Baseline | 57.98% ± 7.31 | 60.11% ± 3.79 | 14.76% ± 2.50 | 24.58% ± 15.00 |
| DeepLab CowMix | 72.81% ± 8.10 | 74.37% ± 4.38 | 19.83% ± 2.28 | 28.86% ± 17.15 |
| DeepLab Full | 84.97% ± 0.46 | 87.92% ± 0.42 | 34.19% ± 0.94 | 81.80% ± 0.78 |
| U-net Baseline | 28.57% ± 6.63 | 40.27% ± 6.37 | 9.18% ± 3.09 | 7.68% ± 5.95 |
| U-net CowMix | 60.13% ± 10.67 | 49.20% ± 19.50 | 20.10% ± 2.95 | 13.99% ± 13.67 |
| U-net Full | 76.83% ± 3.57 | 83.25% ± 0.81 | 30.99% ± 2.60 | 58.35% ± 8.13 |

| | Bus | Car | Motorbike | Train |
|---|---|---|---|---|
| Adv. Baseline | 9.60% ± 10.59 | 53.80% ± 6.91 | 37.40% ± 5.57 | 46.60% ± 5.99 |
| Adv. Semi-sup. | 1.60% ± 3.20 | 58.40% ± 16.80 | 38.60% ± 26.44 | 68.40% ± 2.94 |
| Adv. Full | 51.60% ± 1.20 | 81.00% ± 0.63 | 78.20% ± 0.75 | 79.40% ± 0.49 |
| DeepLab Baseline | 15.86% ± 10.62 | 49.61% ± 6.59 | 36.08% ± 5.83 | 52.30% ± 6.91 |
| DeepLab CowMix | 25.41% ± 14.08 | 67.77% ± 5.12 | 52.66% ± 10.88 | 67.04% ± 3.32 |
| DeepLab Full | 52.38% ± 1.46 | 81.93% ± 0.46 | 80.22% ± 0.61 | 79.96% ± 0.46 |
| U-net Baseline | 4.00% ± 4.00 | 33.70% ± 3.04 | 13.26% ± 6.16 | 24.99% ± 8.00 |
| U-net CowMix | 14.15% ± 8.28 | 47.94% ± 10.07 | 27.13% ± 8.12 | 57.67% ± 4.95 |
| U-net Full | 36.78% ± 11.81 | 73.91% ± 1.66 | 64.44% ± 3.96 | 73.71% ± 0.78 |

| | Bottle | Chair | Dining table | Potted plant |
|---|---|---|---|---|
| Adv. Baseline | 68.80% ± 0.75 | 4.20% ± 5.38 | 29.60% ± 24.42 | 8.00% ± 7.67 |
| Adv. Semi-sup. | 79.40% ± 0.80 | 0.60% ± 0.80 | 33.00% ± 28.93 | 1.00% ± 1.10 |
| Adv. Full | 83.00% ± 0.00 | 57.40% ± 0.80 | 79.80% ± 0.75 | 43.60% ± 1.62 |
| DeepLab Baseline | 70.93% ± 2.15 | 9.44% ± 9.85 | 28.80% ± 23.68 | 12.51% ± 9.71 |
| DeepLab CowMix | 78.10% ± 0.39 | 14.53% ± 17.63 | 32.57% ± 28.15 | 16.88% ± 15.62 |
| DeepLab Full | 84.45% ± 0.11 | 56.66% ± 1.03 | 82.06% ± 0.46 | 46.21% ± 1.11 |
| U-net Baseline | 61.96% ± 1.97 | 7.75% ± 8.12 | 13.44% ± 11.28 | 5.69% ± 5.38 |
| U-net CowMix | 75.26% ± 2.51 | 12.34% ± 10.32 | 17.43% ± 14.52 | 12.55% ± 12.38 |
| U-net Full | 82.71% ± 0.54 | 53.68% ± 5.34 | 70.51% ± 2.05 | 40.19% ± 2.09 |

| | Sofa | TV/monitor |
|---|---|---|
| Adv. Baseline | 34.40% ± 28.58 | 39.80% ± 8.13 |
| Adv. Semi-sup. | 35.00% ± 29.45 | 18.20% ± 12.51 |
| Adv. Full | 82.20% ± 0.75 | 64.60% ± 2.50 |
| DeepLab Baseline | 34.95% ± 28.67 | 43.72% ± 9.62 |
| DeepLab CowMix | 39.51% ± 32.95 | 63.18% ± 1.10 |
| DeepLab Full | 81.98% ± 0.13 | 68.15% ± 1.80 |
| U-net Baseline | 17.64% ± 16.75 | 29.58% ± 7.32 |
| U-net CowMix | 33.16% ± 27.70 | 46.93% ± 6.07 |
| U-net Full | 75.06% ± 1.67 | 70.86% ± 1.11 |

Table 6: Per-class performance on augmented PASCAL VOC dataset, 100 supervised samples

