# OpenReview forum: "Semi-supervised semantic segmentation needs strong, high-dimensional perturbations"
_ICLR.cc/2020/Conference — Reject_

### Official Review · AnonReviewer1 · 2019-10-21
**Official Blind Review #1**

**Rating:** 3

**Review:**

This work analyzes the consistency regularization in semi-supervised semantic segmentation. Based on the results on a toy dataset, this work proposes a novel regularization for semi-supervised semantic segmentation, which is named CowMix.

Pros:
-- The proposed CoxMix is easy to understand and implement.
-- The experimental results seem to benefit from this proposed CoxMix at a first glance.

Cons:
The writing is not clear. Sometimes I have to make a ``guess" about the technical details. For example:
-- Other than L_{cons}, is there any other Loss term utilized in this work? Based on Figure 3, it seems only L_{cons} is utilized. If so, is it a waste not to use the label training data (although very few) to calculate a cross-entropy loss?

-- It seems the experimental setting in this submission follows the settings in Hung, 2018. However, for the experiment on VOC 2012 validation set, Hung tested their method on 1/8 1/4 1/2 of labeled data (Table 1). While in this submission, Table 3 shows the results on label data of 100, 200, 400, 800, 2646(25%). The split ratios seem different from Hung's work, which confuses me.

-- "Note that in context of semantic segmentation, all geometric transformations need to be applied in reverse for the result image before computing the loss (Ji et al., 2018). As such, translation turns into a no-op, unlike in classification tasks where it remains a useful perturbation."
    Is there any experimental result to support this claim?

-- It is a little hard for me to fully understand Figure 2. For example, how to get 2. (c)? What is the meaning of the word "gap" here?


**Experience Assessment:**

I have read many papers in this area.

**Review Assessment: Checking Correctness Of Derivations And Theory:**

I assessed the sensibility of the derivations and theory.

**Review Assessment: Checking Correctness Of Experiments:**

I assessed the sensibility of the experiments.

**Review Assessment: Thoroughness In Paper Reading:**

I read the paper at least twice and used my best judgement in assessing the paper.

---

> ### Author Response · Authors · 2019-11-11
> **Response to Official Blind Review #1**
>
> Thank you for your review. We appreciate your effort and we thank you for highlight areas in need of clarification.
> If there are any others that come to mind, please feel free to let us know.
>
> We now state explicitly that we do use supervised loss in section 4.1 paragraph 2. In order to stay within paper
> length limitations, we removed some details from this paragraph as they are also present in Appendix D.3. We
> will look further at Figure 3 before the end of the response/discussion period.
>
> We used different split ratios to that of Hung et al. as the practical benefit of semi-supevised learning is maximised by
> reducing the required number of labelled samples -- and therefore the effort required to label them -- by
> as much as possible. We therefore tested our approach using a significantly smaller number of labelled samples
> in order to illustrate that our approach gives strong performance in these challenging but practically useful
> conditions. Labelling 1,323 images (12.5% of augmented Pascal) or 5,291 images (50%) requires a considerable
> amount of manual labour.
>
> We have attempted to answer your query concerning applying geometric transformations in reverse in Appendix D.1.1.
> In short, classification is translation invariant while semantic segmentation is translation *variant*. As a consequence
> in semantic segmentation scenarios any translation must in effect be reversed elsewhere in the pipeline to prevent
> the network from learning from erroneous training data. That said, using translation can cause a part of the image
> to take a slightly different path through the convolutional layers of the network, so it can provide a small
> improvement. Our standard augmentation based unsupervised regularizer can and does utilise this, although it does not
> achieve gains in semi-supervised segmentation.
>
> Figure 2: we have replaced with word gap with the term 'low density region' that is consistent with the rest of the
> paper. We constructed two similar artificial scenarios for Fig 2 (a) and (b); one with a low density region
> separating the two regions of unsupervised samples and one without.
>
> We have expanded the text in Appendix A to explain how Figure 2(c) was made. We hope this clears things up.
> If not, please feel free to let us know.

---

### Official Review · AnonReviewer2 · 2019-10-23
**Official Blind Review #2**

**Rating:** 3

**Review:**

This paper provided first provided analysis for the problem of semantic segmentation. Through a few simple example, the authors suggested that the cluster assumption doesn’t hold for semantic segmentation. The paper also illustrated how to perturb the training examples so that consistency regularization still works for semantic segmentation.
The paper also introduce a perturbation method that can achieve high dimensional perturbation, which achieve solid experimental results.

The analysis part seems interesting and innovative to me. But it is very qualitative and I'm not fully convinced that the analysis on 2d example can actually carry over to high dimensional spaces for images. I also don't quite see the connection between the toy example and the proposed perturbation method. For example, why the proposed perturbation method has the property of "the probability of a perturbation crossing the true class boundary must be very small compared to the amount of exploration in other dimensions"?

The proposed algorithm is an extension of the existing cutout and cut mix. The way to generate new mask is a very smart design to me. This should be the most important contribution of the paper.

The writing of the paper is very clear and easy to follow. The experimental results look very convincing overall and proposed algorithm does show very promising results.

To sum up, the paper is an ok paper from the practical perspective, but the analysis in the paper wasn't strong enough to me.

**Experience Assessment:**

I do not know much about this area.

**Review Assessment: Checking Correctness Of Derivations And Theory:**

I assessed the sensibility of the derivations and theory.

**Review Assessment: Checking Correctness Of Experiments:**

I assessed the sensibility of the experiments.

**Review Assessment: Thoroughness In Paper Reading:**

I read the paper at least twice and used my best judgement in assessing the paper.

---

> ### Author Response · Authors · 2019-11-11
> **Response to Official Blind Review #2**
>
> Thank you for your review and thank you for identifying areas that are of concern.
>
> I will attempt to answer your query concerning how our analysis of a 2D example carries over to a high dimensional
> problem.
>
> To recap, Figure 2(d) shows that consistency regularization can succeed without requiring low density regions in the input
> distribution by constraining the perturbations that drive consistency regularisation to be parallel to the
> intended decision boundary. In such a simple 2D example, perturbing along a line parallel to the decision boundary
> (or rather parallel to a line tangent to the decision boundary at the closest point on said boundary)
> is sufficient. In higher dimensions, perturbing a sample in one direction (making it trace out a line) is insufficient as
> the decision boundary is free to orient itself almost arbitrarily while still being perpendicular to the line of perturbation.
> In order to properly constrain the orientation of the decision boundary, the perturbations must operate in as many
> dimensions as possible. The cutting and mixing regularizers (CutOut, CutMix, CowOut and CowMix) discussed in this paper
> are high dimensional as an axis of perturbation is supplied by each pixel in the cutting/mixing mask.
>
> Furthermore, masking out part of an object (as in CutOut or CowOut) does not change the ground truth class of the pixels
> of the object that remain, hence these perturbations do not cross the class boundary. Mixing the images does not change
> the class for a similar reason.
>
> It is our intention look carefully at the wording of this part of the paper in the next few days.

---

> > ### Author Response · Authors · 2019-11-15
> > **Response to Official Blind Review #2 (2)**
> >
> > We have added a little more to Section 3.2 paragraph 4, in that we have stated that the perturbations should be high dimensional in order to adequately constrain a decision boundary in the high-dimensional space of natural images.

---

### Official Review · AnonReviewer3 · 2019-10-23
**Official Blind Review #3**

**Rating:** 3

**Review:**



# Summary

This paper proposes a method for semi-supervised semantic segmentation.
The authors tackle this problem through consistency regularization, a successful technique in image classification, that encourages the network to give consistent predictions on unlabeled samples which are perturbed in multiple ways. The authors argue that the cluster assumption (to which effectiveness of consistency regularization has been partially attributed) does not hold in semantic segmentation. Thus, in order to enable class boundaries to become low-density regions and then better guide contrastive regularization, the authors argue that a stronger perturbation must be inserted. To this effect they first propose looking at CutOut and CutMix types of methods. They improved upon them by putting forward a variant of CutMix, coined CowMix, with more degrees of freedom and using flexible masks instead of rectangular ones. CowMix is evaluated on the Cityscapes and PascalVOC 2012 datasets in the semi-supervised regime and showing encouraging results.


# Rating
I find the paper and the advanced ideas of interest for the community and I consider they are novel. I'm currently on the fence between Weak Accept and Weak Reject, mostly due to incomplete evaluations and support for claims made in the introduction regarding the infeasibility of contrastive regularization methods for semantic segmentation. I would be happy to upgrade my rating if authors addressed these concerns.


# Strong points
- The paper is well written and mostly clear with a good coverage and positioning w.r.t. related work. The authors illustrate well the reasoning and the choices they have made. The author provide plenty of ablation studies (e.g., per class statistics) and implementation details, improving significantly the reproducibility of the contribution.
- The flexible masking technique that is advanced here is novel and experimentally seems effective.
- This work is among the few that address semi-supervised semantic segmentation in a non-adversarial manner, so I would give it some novelty credit.
- I appreciate the evaluation protocol of averaging across multiple runs.

# Weak points

## Unclear aspects
- The authors argue that consistency regularization has had little success so far in semantic segmentation problems since low density regions in input data do not align well with class boundaries. It would be useful to provide a reference to this claim or at least validate it experimentally on a large dataset.

- In Figure 1, it is not clear on which features where the distances between patches computed? Is it on raw pixels or intermediate feature maps from a CNN?
If the distances are made over raw pixels, I find it difficult to make the connection between distances in the pixel space and distances in the class space.
Are the neighbor patches overlapping with the central/query patch?

## Experiments
- The authors compare against other methods on the CamVid dataset. CamVid is a small and relatively limited dataset (~367 images for training from the streets of Cambridge). I'm worried that this dataset might not be enough to conclude and emphasize the benefits of this method over other semi-supervised techniques. For instance CowOut does not seem do be above CutOut, while CutMix has convergence problems and low scores.
The other experiments on Cityscapes and Pascal VOC are certainly interesting, but the method is compared only against Hung et al. which a different family of methods and the subset baseline (which is useful but not enough). I think this work would benefit from an additional baseline in the style of contrastive regularization methods, e.g. ICT, and eventually CutOut, to support the initial arguments regarding the limitations of these methods in semantic segmentation and respectively the effectiveness of the flexible masks over the rectangular ones in this setup.


# Suggestions for improving the paper:
1) It would be useful to include other semi-supervised baselines, e.g. ICT, and the baseline perturbation CutMix on larger experiments, in order to better emphasize the contributions of this work.

2) Did the authors try the flexible masking on image classification? How is it expected to perform over ICT, MixUp or MixMatch?


**Experience Assessment:**

I have published one or two papers in this area.

**Review Assessment: Checking Correctness Of Derivations And Theory:**

I assessed the sensibility of the derivations and theory.

**Review Assessment: Checking Correctness Of Experiments:**

I carefully checked the experiments.

**Review Assessment: Thoroughness In Paper Reading:**

I read the paper at least twice and used my best judgement in assessing the paper.

---

> ### Author Response · Authors · 2019-11-11
> **Response to Official Blind Review #3**
>
> Thank you for your very helpful and detailed review. You have identified several areas in which we can improve and clarify our work.
>
> We have updated the caption in Figure 1 and the text in Section 3.1 to clarify that we are comparing the contents
> of overlapping patches that are centred on neighbouring pixels; a patch A is compared with a neighbouring patch B
> that is shifted e.g. one pixel to the right of patch A. Figure 1 (b) and (c) are generated by computing the distance between patches that are centered on each pixel in the image.
>
> We have expanded Section 3.1 to confirm that we are compare the raw pixel content of image patches. While the use
> of raw pixel space -- as opposed to e.g. a feature space from a pre-trained network -- may seem unusual given the
> highly non-linear nature of neural networks, prior consistency regularisation based semi-supervised learning approaches
> (e.g. Laine et al. etc.) apply sample perturbation in the input space. We specifically reference the virtual
> adversarial approach of Miyato et al. as they use adversarial techniques to generate an adversarial example
> $\hat{x}$ from $x$ that maxmimizes the distance between predictions $d(\hat{y}, y)$ (where y = f_\theta(x)).
> Once again this perturbation is performed in raw pixel / input space, illustrating the significance of the input space
> data distribution.
>
> As far as we know the data distribution of semantic segmentation problems has not been studied in prior work.
> To recap, we observed that computing the distance between overlapping neighbouring patches is equivalent to applying
> a uniform filter to the squared gradient image, thus suppressing the fine details that would be required for
> low density regions to manifest along object or texture boundaries. We believe that this should apply to natural
> images in general. We have however analysed the patch distribution in the Cityscapes dataset to confirm this.
> Due to the space limitations we have added this as Appendix B.
>
> We are currently in the process of conducting experiments on the Cityscapes dataset using CutOut, CutMix
> and CowOut using the DeepLab2 network. We hope to get results for ICT as well. We will revise the paper
> later during the rebuttal period to include them when the experiments have finished running. We will only be able
> to do this for 372 supervised samples prior to the end of the rebuttal period due to lack of available
> compute at this time and the short amount of time available. We will endeavour to produce a complete
> set of results should our work be accepted.
>
> Applying CowOut and CowMix to classification problems is a topic that we intend to explore in further work.
> We believe that the challenging nature of semi-supervised semantic segmentation that we have described
> in this work has resulted in a regularizer that could have interesting properties when used in classification
> and are eager to evaluate it.

---

> > ### Author Response · Authors · 2019-11-15
> > **Response to Official Blind Review #3 (2)**
> >
> > We have added results for ICT, CutOut, CutMix and CowOut using DeepLab2 for Cityscapes with 372 supervises samples. We will run experiments to produce results for other values for number of supervised samples, U-Net architecture and the Pascal dataset in due course.
> >
> > The new results show that CutOut harms performance, CutMix contributes an improvement (but only just) while CowOut makes a fair improvement, but trailing that of CowMix, strengthening the position of CowOut and CowMix relative to the others.

---

### Decision · Program_Chairs · 2019-12-19

**Decision:**

Reject

**Comment:**

This paper proposes a method for semi-supervised semantic segmentation through consistency (with respect to various perturbations) regularization. While the reviewers believe that this paper contains interesting ideas and that it has been substantially improved from its original form, it is not yet ready for acceptance to ICLR-2020. With a little bit of polish, this paper is likely to be accepted at another venue.